# Deletion of Wnt10a Is Implicated in Hippocampal Neurodegeneration in Mice

**DOI:** 10.3390/biomedicines10071500

**Published:** 2022-06-25

**Authors:** Jia-He Zhang, Takashi Tasaki, Manabu Tsukamoto, Ke-Yong Wang, Kin-ya Kubo, Kagaku Azuma

**Affiliations:** 1Department of Anatomy, School of Medicine, University of Occupational and Environmental Health, 1-1 Iseigaoka, Yahatanishi-ku, Kitakyusyu 807-8555, Fukuoka, Japan; jiahe@med.uoeh-u.ac.jp; 2Department of Pathology, Kagoshima University Graduate School of Medical and Dental Sciences, 8-35-1 Sakuragaoka, Kagoshima 890-8544, Kanagawa, Japan; t-tasaki@m2.kufm.kagoshima-u.ac.jp; 3Department of Orthopedic Surgery, School of Medicine, University of Occupational and Environmental Health, 1-1 Iseigaoka, Yahatanishi-ku, Kitakyusyu 807-8555, Fukuoka, Japan; m-tsuka@med.uoeh-u.ac.jp; 4Shared-Use Research Center, School of Medicine, University of Occupational and Environmental Health, 1-1 Iseigaoka, Yahatanishi-ku, Kitakyusyu 807-8555, Fukuoka, Japan; kywang@med.uoeh-u.ac.jp; 5Faculty of Human Life and Environmental Science, Nagoya Women’s University, 3-40 Shioji-cho, Mizuho-ku, Nagoya 467-8610, Aichi, Japan; kubo@nagoya-wu.ac.jp

**Keywords:** hippocampus, Wnt10a, neurogenesis, neuroinflammation, spatial memory, synapse

## Abstract

The hippocampus plays an important role in maintaining normal cognitive function and is closely associated with the neuropathogenesis of dementia. Wnt signaling is relevant to neuronal development and maturation, synaptic formation, and plasticity. The role of Wnt10a in hippocampus-associated cognition, however, is largely unclear. Here, we examined the morphological and functional alterations in the hippocampus of Wnt10a-knockout (Wnt10a^-/-^) mice. Neurobehavioral tests revealed that Wnt10a^-/-^ mice exhibited spatial memory impairment and anxiety-like behavior. Immunostaining and Western blot findings showed that the protein expressions of β-catenin, brain-derived neurotrophic factor, and doublecortin were significantly decreased and that the number of activated microglia increased, accompanied by amyloid-β accumulation, synaptic dysfunction, and microglia-associated neuroinflammation in the hippocampi of Wnt10a^-/-^ mice. Our findings revealed that the deletion of Wnt10a decreased neurogenesis, impaired synaptic function, and induced hippocampal neuroinflammation, eventually leading to hippocampal neurodegeneration and memory deficit, possibly through the β-catenin signaling pathway, providing a novel insight into preventive approaches for hippocampus-dependent cognitive impairment.

## 1. Introduction

Alzheimer’s disease (AD), the most common type of dementia, is an irreversible neurodegenerative disorder affecting millions of people in the world. It is becoming a major global public health issue. The progressive memory deficit and cognitive impairment in AD patients are associated with alterations in hippocampal morphology and function, including neurogenesis, synaptic plasticity, and microglia-associated neuroprotection. The hippocampus is composed of the cornu ammonis (CA), dendate gyrus (DG), and subiculum, which play important roles in maintaining normal learning and memory [1]. It is also closely associated with the pathogenesis of dementia and many other neuropsychological disorders. The hippocampal DG has a remarkable capacity to generate new neurons throughout life. Hippocampal neurodegeneration is involved in spatial learning and memory and is strongly correlated with cognitive impairment [2]. Internal and external factors, such as psychological stress, enriched environment, and gene mutations, affect hippocampal neurogenesis [3,4,5]. The microglia are the main innate immune cells in the brain and play a critical role in the pathological process of AD.

Dysregulated microglia release various pro-inflammatory cytokines and exacerbate amyloid-β accumulation, synaptic dysfunction, and cognitive deficit in AD progression [6].

The Wnt family of 19 secreted glycoproteins has a critical role in regulating diverse biological processes, including embryonic development, cell differentiation, and adult tissue maintenance through canonical Wnt/β-catenin and noncanonical pathways. Growing evidence has revealed the function of the Wnt/β-catenin signaling pathway in neuronal development and maturation, synaptic formation, and plasticity. It was reported that the Wnt/β-catenin signaling pathway plays a vital role in numerous cellular and molecular processes for maintaining higher function of the brain [7]. The dysregulation of Wnt/β-catenin signaling was linked to hippocampus-dependent cognitive impairment and AD [8]. Wnt signaling has been implicated in hippocampal neurogenesis. The blockage of Wnt signaling decreased hippocampal neurogenesis in vitro and in vivo [9]. Wnt blockage also induced synaptic degeneration [10]. Brain-derived neurotrophic factor (BDNF) plays a crucial role in neurogenesis and synaptic plasticity. Wnt signaling was reported to induce BDNF expression in neurons and glia [11]. The downregulation of the Wnt/β-catenin signaling pathway was associated with increased nuclear factor-κB (NF-κB) signaling and was involved in neuroinflammation [12]. An impaired Wnt signaling pathway is associated with enhanced neuroinflammation and increased amyloid-β aggregation. Dysfunctional Wnt signaling might be a key event contributing to the pathogenesis of AD [13].

Wnt10a mutations are related to the development of ectodermal dysplasia, including dental defects, nail dystrophy, and thinning hair [14]. A typical disease caused by Wnt10a deficiency is odonto-onycho-dermal dysplasia, which has symptoms typically affecting skin, hair, and teeth [15]. Recent studies have indicated that Wnt10a plays an essential role in tumor growth, wound healing, osteogenesis, adipogenesis, and female fertility using Wnt10a^-/-^ mice [16,17,18,19]. No reports, however, have examined the effects of Wnt10a on hippocampal structure and function. In the present study, we aim to explore the potential involvement of Wnt10a-mediated hippocampal functional morphology using Wnt10a^-/-^ mice. Our findings demonstrate that Wnt10a plays a critical role in maintaining normal hippocampal morphology and function in mice.

## 2. Materials and Methods

### 2.1. Animals

Wnt10a-knockout (Wnt10a^-/-^) mice were generated as described previously [18]. Briefly, we obtained embryonic stem cells carrying the deleted 12,663 base pairs of mouse chromosome 1, which included the entire Wnt10a coding region, from the University of California, Davis, School of Veterinary Medicine (Davis, CA, USA). The embryonic stem cell clones were microinjected into the blastocysts of C57BL/6J mice to generate chimeric mice. The successful deletion of Wnt10a was determined with a genotyping PCR method. The experiment was performed on 20- to 25-week-old male Wnt10a^-/-^ mice and age-matched male wild-type C57BL/6J (WT) mice. All the mice were provided with standard rodent diet (CE-2, CLEA Japan, Inc., Tokyo, Japan) and drinking water ad libitum under controlled temperature (23 ± 1 °C), humidity (55 ± 5%), and a 12 h light:dark cycle. The experimental protocol was approved by the Ethics Committee for Animal Care and Experimentation of the University of Occupational and Environmental Health, Japan (AE18-029, 14 February 2019).

### 2.2. Neurobehavioral Tests

Neurobehavioral tests were conducted in the Animal Behavior Facility at the University of Occupational and Environmental Health during the light phase (between 09:00–13:00). Spatial memory was evaluated using the Barnes maze test, a standardized method for assessing spatial memory in rats and mice [20]. The utilized maze comprised a circular platform (92 cm in diameter) with 20 equally spaced holes around the periphery. One hole was connected to a hidden escape cage. Three visual cues of different shapes and colors were provided during the maze test. The animals were allowed to acclimatize to the testing room for 1 h. Then, the mice (*n* = 5/group) were placed in the center of the platform to explore freely. If a mouse did not locate the hidden escape cage within 3 min, it was guided manually to the escape hole. After each trial, the maze and escape cage were cleaned with 70% ethanol. All the mice were given 2 trials every day for 7 consecutive days. All the trials were recorded with a video camera installed over the platform, and the data were analyzed using Ethovision XT 15 software (Noldus Information Technology, Leesburg, VA, USA). The escape latency, velocity, number of errors to the target hole, and distance moved to the target hole were calculated for evaluating the spatial memory.

An open field test was used to evaluate anxiety-like behavior [21]. The open field apparatus was a square chamber of 40 × 40 cm. The central area was defined as 20 × 20 cm in the middle of the arena. One hour before the start of testing, the animals were brought into the behavioral testing room for adaption. The mice (*n* = 5/group) were gently placed in the corner of the box. Mouse behavior was video-recorded for 20 min and analyzed with Ethovision XT 15 software (Noldus Information Technology). We monitored and recorded the total distance moved, the number of entries into the central area, and the time spent in the center of the open field. After each trial, the apparatus was cleaned with 70% ethanol.

### 2.3. Histological Observation

The mouse brains (*n* = 11/group) were carefully extracted and fixed in 10% neutral formalin solution or 4% paraformaldehyde solution for 24 h at 4 °C. After gradient dehydration and paraffin embedding, the brain coronary serial sections (5 μm thickness) were prepared, stained with hematoxylin-eosin, and examined with a light microscope (BX50, Olympus Corporation, Tokyo, Japan). The contours of the left and right hippocampi were traced, and the cross-sectional areas were measured using imaging software (CellSens, Olympus Corporation). The hippocampal volume was determined by multiplying the total area by the section thickness (5 μm) and the sampling interval (30), as reported previously [22].

### 2.4. Immunostaining

Immunohistochemical staining was performed as described previously [23]. Paraffin sections were routinely deparaffinized in xylene and rehydrated in ethanol. The sections were soaked in antigen retrieval solution (Dako, Santa Clara, CA, USA) with an autoclave at 121 °C for 15 min. After treatment with Protein Block Serum-Free (Dako), the sections were incubated with anti-Iba1 (Fujifilm, Tokyo, Japan, Table 1) at 4 °C overnight. The sections were then incubated with biotinylated goat anti-rabbit IgG and a streptavidin peroxidase complex (Nichirei Biosciences Inc., Tokyo, Japan) for 30 min. The sections were then treated with diaminobenzidine and counterstained with hematoxylin. The sections were examined and recorded using a light microscope (VS120, Olympus Corporation) connected to a digital camera. Three sections per animal and four fields per section were counted with an original magnification of ×200. The average number of Iba1-positive microglia in the unit area was calculated (*n* = 5/group).

For immunofluorescence staining, the sections were pretreated with antigen retrieval solution and incubated with anti-doublecortin (Abcam, Cambridge, UK), anti-BDNF (Abcam), anti-β-catenin (Abcam), and anti-Wnt10a (Table 1) at 4 °C overnight. The anti-Wnt10a antibody was produced in rabbits immunized with a synthetic peptide corresponding to amino acids 160–172 of mouse Wnt10a [18]. The sections were then treated with Alexa Fluor 488-conjugated anti-rabbit IgG (1:200, Thermo Fisher Scientific, Waltham, MA, USA) and DAPI (0.1 μg/mL, Thermo Fisher Scientific) to detect fluorescence and nuclei. Images were captured with a light microscope (VS120, Olympus Corporation) and digitized with OlyVIA software (Olympus Corporation). For the quantification of neurogenesis in the hippocampal DG, the number of doublecortin-positive cells was counted in 5 sections per animal (*n* = 5/group). A total of 10 randomly selected fields per section (original magnification of ×400) were calculated [24]. All the measurements were conducted in a randomized, double-blind procedure under the same conditions.

### 2.5. Transmission Electron Microscopy

The mouse brains was carefully dissected, and the hippocampal CA1 regions were harvested under a stereo microscope (*n* = 5/group). The tissues were fixed in 2.5% glutaraldehyde in 0.1 M phosphate buffer (pH 7.4) overnight at 4 °C and postfixed in 1% OsO_4_ for 60 min. After dehydrating through an ascending, graded acetone series, the specimens were embedded in epoxy resin. Ultrathin sections were prepared with a diamond knife using an ultramicrotome (EM UC7, Leica, Wetzlar, Germany). After staining with uranyl acetate and lead salts, the sections were examined using a transmission electron microscope (JEM-1400Plus, JEOL Ltd., Tokyo, Japan). Synapses were identified by the existence of synaptic vesicles and postsynaptic density (PSD). The PSD length was measured as previously described [25]. Fifty synapses were determined per animal.

### 2.6. Western Blot Analyses

The mouse hippocampi were lysed with radioimmunoprecipitation assay buffer (Millipore, Burlington, MA, USA) and centrifuged at 12,000 rpm for 30 min at 4 °C, and the supernatants were collected (*n* = 5/group). The hippocampal protein concentration was evaluated using a BCA protein assay kit (Thermo Fisher Scientific). The protein was adjusted to 2.0 μg/μL, isolated in 4–12% Bis-Tris Gel (Thermo Fisher Scientific), and transferred to PVDF membranes (Millipore, Bedford, MA, USA). After blocking with 1% skim milk, immunoblotting was performed with the primary antibodies summarized in Table 1. After rinsing with buffer, the immunoblotted membranes were incubated with the corresponding secondary antibodies (1:3000, Cell Signaling Technology, Danvers, MA, USA) for 60 min, and the signals were detected with an ECL kit (Cytiva, Buckinghamshire, UK). The target protein band was analyzed using an Ez-Capture MG system (Atto Corp., Tokyo, Japan), and the densitometric analysis of the bands was performed with Scion Image software (version 4.0.2, Scion Corp., Frederick, MD, USA). GAPDH was used as an internal reference to normalize the target protein expression.

### 2.7. Statistical Analyses

Statistical analyses were performed using GraphPad Prism version 7.03 (GraphPad Software, San Diego, CA, USA). All the data were expressed as mean ± SEM. The cumulative incidence of latency was analyzed using a log-rank (Mantel–Cox) test, and the hazard ratio was estimated with the Mantel–Haenszel method [26]. The other parameters of the Barnes maze test were analyzed using two-way repeated measure ANOVA, including the effects of group and time. When significant differences were indicated, they were further analyzed using Fisher’s least significant difference test. The open field behavioral data, as well as the histological and Western blot analyses, were evaluated using Student’s *t*-test for the comparison of two groups. A value of *p* < 0.05 was considered to be statistically significant.

## 3. Results

### 3.1. Wnt10a^-/-^ Mice were Smaller in Size and had Lower Hippocampal Volume

Wnt10a^-/-^ mice were grossly smaller in size when compared with WT mice of the same age (Figure 1A). The mean body weights and brain weights were significantly lower in Wnt10a^-/-^ mice (Figure 1B,C), corresponding to previous reports [17,18,19]. Compared with WT mice, the hippocampal volume in Wnt10a^-/-^ mice was significantly lower (Figure 1D,E). When adjusting the hippocampal volumes for body weight, there was no significant difference between WT and Wnt10a^-/-^ mice.

### 3.2. Wnt10a^-/-^ Mice Showed Impaired Spatial Memory and Anxiety-like Behavior

As shown in Figure 2A, the cumulative incidence of escape in Wnt10a^-/-^ mice was longer than that for the WT mice. The hazard ratio [95%-CI] of Wnt10a^-/-^ vs. WT mice was 0.42 [0.24 to 0.72] (*p* = 0.0017). Wnt10a^-/-^ mice had a rate to reach the escape hole decreased by 58% from that of WT mice. The two-way ANOVA showed a significant group effect (*p* < 0.0001) and training day effect (*p* = 0.0001) in the number of errors to the target hole, with no interaction between group and training day (*p* = 0.0804). All the animals showed improved performance during the 7 training days. The number of errors to the target hole was significantly higher in Wnt10a^-/-^ mice compared with WT mice, indicating spatial learning and memory impairment in the Wnt10a^-/-^ mice (Figure 2B). The distance moved to the target hole was increased in Wnt10a^-/-^ mice compared with WT mice (Figure 2C,D). The training day effect on the distance moved to the target hole was statistically significant (*p* = 0.0006). There were no interactions between group and training day regarding the distance moved to the target hole (*p* = 0.2380). The findings of the open field test showed that the total distance moved, the number of entries into the central area, and the time spent in center of the box were significantly decreased in Wnt10a^-/-^ mice compared with WT mice (Figure 2E–H), indicating anxiety-like behavior.

### 3.3. Wnt10a^-/-^ Mice Exhibited Lower Expression of β-Catenin in the Hippocampus

We initially examined the expression of Wnt10a and β-catenin signaling in the hippocampus by immunostaining and Western blot analysis. Wnt10a-positive cells and protein expression were present in the hippocampi of WT mice. There were no Wnt10a-positive cells, nor protein expression, in the hippocampi of Wnt10a^-/-^ mice (Figure 3A,C). Compared with WT mice, β-catenin-positive cells and protein expression were significantly decreased in the hippocampi of Wnt10a^-/-^ mice (Figure 3B,D).

### 3.4. Wnt10a^-/-^ Mice Displayed Lower Neurogenesis and BDNF Expression in the Hippocampus

Neurogenesis in the hippocampal DG was evaluated by analyzing the expression level of doublecortin, an indicator for adult neurogenesis. The average number of doublecortin-positive cells of the hippocampal DG was significantly lower in Wnt10a^-/-^ mice than in WT mice (Figure 4A,C). Immunostaining and Western blot analysis indicated significantly lower expressions of BDNF in the hippocampi of Wnt10a^-/-^ mice compared with WT mice (Figure 4B,D).

### 3.5. Wnt10a^-/-^ Mice Showed Synaptic Dysfunction in the Hippocampus

We examined hippocampal synapses using electron microscopy. Compared with WT mice, the presynaptic vesicles were decreased, and the PSD length was significantly shorter in Wnt10a^-/-^ mice (Figure 5A). We also evaluated the protein expression of hippocampal PSD95 and synaptophysin with Western blot analysis. The results showed that both PSD95 and synaptophysin expression levels were significantly lower in the hippocampi of Wnt10a^-/-^ mice compared with WT mice (Figure 5B), suggesting synaptic dysfunction.

### 3.6. Wnt10a^-/-^ Mice Exhibited Neuroinflammation in the Hippocampus

Immunostaining was carried out by ionizing calcium-binding adapter molecule 1 (Iba1) as a microglial marker. We observed that the Iba1-positve microglia levels were significantly increased in the hippocampi of Wnt10a^-/-^ mice (Figure 6A,C). We next examined the ultrastructural features of the hippocampal microglia using electron microscopy. Increased numbers of lysosomes and dark granular bodies were seen in the hippocampal microglial cytoplasm of Wnt10a^-/-^ mice (Figure 6B). We also determined the protein expression of hippocampal Iba1. The protein expression level of the hippocampal Iba1 was significantly higher in Wnt10a^-/-^ mice than in WT mice (Figure 6D). Western blot analysis showed that the protein expression level of amyloid-β in the hippocampi of Wnt10a^-/-^ mice was significantly higher than that in WT mice (Figure 6E), reflecting the accumulation of amyloid-β plaques in the hippocampus. The hippocampal protein expressions of NF-κB, glycogen synthase kinase 3β (GSK-3β), TNF-α, and IL-1β were significantly elevated in Wnt10a^-/-^ mice, indicating hippocampal neuroinflammation (Figure 6E).

## 4. Discussion

In the present study, we confirmed that Wnt10a^-/-^ mice showed hippocampus-dependent spatial memory impairment and anxiety-like behavior. Both the Barnes maze and the Morris water maze can evaluate the spatial memory of rats and mice. The Morris water maze is considered to be stressful because of forced swimming. Neurobehavioral tests involving strong stress can affect animal performance. Therefore, the Barnes maze is a more ideal task for eliminating confounds induced by psychological stress [27,28,29]. The escape latency was significantly longer, and the number of errors to the target hole and the distance moved were increased while the speed was decreased, suggesting spatial memory impairment in Wnt10a^-/-^ mice. The results of the open field test exhibited a significant decrease in the total distance moved, the number of entries into the central area, and the time spent in the center of the box in Wnt10a^-/-^ mice, indicating anxiety-like behavior, as reported previously [30].

Adult hippocampal neurogenesis is closely associated with spatial learning and memory, as well as anxiety-like behavior, by modulating information processing within the hippocampus. Growing evidence has indicated that adult hippocampal neurogenesis is regulated by Wnt signaling. The decreased expression of Wnt signaling or increased expression of Wnt inhibitors might impair hippocampal neurogenesis [5,7,31,32]. Our findings showed that hippocampal neurogenesis was significantly decreased in Wnt10a^-/-^ mice, corresponding with previous reports [33,34].

BDNF is a major neurotrophin in the brain that contributes to a variety of neuronal activities, including neurogenesis and synaptic plasticity, through its receptor, tropomyosin receptor kinase [35]. Wnt/β-catenin signaling could upregulate BDNF expression [36]. The blockage of Wnt/β-catenin signaling inhibits BDNF expression. The stimulant effect of BDNF on hippocampal neurogenesis was inhibited by a Wnt-signaling-specific blocker [37]. The present study showed that the expression of hippocampal BDNF in Wnt10a^-/-^ mice decreased significantly. We consider that Wnt10a deficiency could cause a reduction in BDNF expression, a decline in hippocampal neurogenesis, dysfunction in synaptic function, and eventually lead to hippocampus-dependent memory deficit.

A synapse is a highly specialized connection between neurons designed to guarantee efficient and precise transmission across neurons. Synaptic structural plasticity reflects physiological function and plays an important role in spatial memory. Synaptic structural and molecular alteration is closely associated with synaptic functional plasticity. Wnt signaling promotes both presynaptic and postsynaptic assembly through different receptors [38]. Physiological studies have demonstrated that Wnt signaling can regulate synaptic transmission and synaptic plasticity [39]. Synaptophysin is a protein localized to the synaptic vesicle, which is regulated by Wnt signaling [40]. The expression level of synaptophysin correlates with cognition [41]. PSD95 is a postsynaptic protein that is essential for synaptic maturation. PSD95 expression is also modulated by the Wnt/β-catenin pathway. The induction of a Wnt antagonist decreased the expression level of PSD95 [42]. In this study, we found, morphologically, that the presynaptic vesicles were decreased, and the postsynaptic densities were shorter and thinner in Wnt10a^-/-^ mice. Molecularly, the protein expression levels of both synaptophysin and PSD95 in the hippocampus were significantly lower in Wnt10a^-/-^ mice, suggesting synaptic dysfunction induced by Wnt10a deficiency.

The accumulation of amyloid-β plaques is a key initiating event in the neuropathogenesis of AD. The activation of Wnt/β-catenin signaling could inhibit the formation of amyloid-β and reduce the deposition of amyloid-β plaques [43,44]. Abnormal amyloid-β accumulation could stimulate microglia recruitment and initiate neuroinflammation, resulting in neuronal degeneration. In this study, we observed increased amyloid-β plaques in the hippocampus, suggesting AD-like pathological features in Wnt10a^-/-^ mice. Microglia play diverse critical roles in modulating neuroinflammation, regulating neuronal activity, and acting as phagocytosis. Microglia can also mediate synaptic loss by engulfing synapses, inducing neuronal degeneration [45,46]. Activated microglia play an important role in neuroinflammation [47].

Wnt signaling affects the cell fate, cell activities, and cell functions of microglia and the associated neuroinflammation. The activation of Wnt/β-catenin signaling prevented microglial activation and alleviated neuroinflammation [48]. Downregulation of the Wnt/β-catenin signaling pathway triggered pro-inflammatory microglial activation, leading to microglia-mediated neuroinflammation [49]. Activated microglia secreted various neurotoxic and inflammatory factors, including IL-1β and TNF-α, ultimately leading to hippocampal neuroinflammation and hippocampus-dependent cognitive impairments [50].

NF-κB is a key regulator for inflammation, which is involved in neuroinflammation and enrichment in the hippocampal neurons [51]. The Wnt/β-catenin pathway regulated inflammatory responses by affecting the expression of NF-κB [52]. Activation in Wnt/β-catenin signaling might decrease the expression of NF-κB [53]. Downregulation of Wnt/β-catenin signaling or upregulation of a Wnt antagonist increased the expression of NF-κB related to neuroinflammation [12].

GSK-3β is a key inhibitor of Wnt/β-catenin signaling involved in inflammation [54]. Downregulation of the Wnt/β-catenin signaling pathway increased the expression of GSK-3β/in mouse hippocampi [55]. Neuroinflammation was associated with a reduced expression in Wnt/β-catenin signaling and an increased expression in GSK-3β [56].The elevated expression of GSK-3β facilitated the accumulation of amyloid-β plaques [23]. In the present study, we observed an increased number of activated microglia and elevated expressions of various inflammatory factors, including NF-κB, TNF-α, and IL-1β, in the hippocampus, reflecting the hippocampal neuroinflammation in Wnt10a^-/-^ mice.

## 5. Conclusions

Our findings provided evidence that Wnt10a^-/-^ mice exhibited spatial memory impairment and anxiety-like behavior. The expression levels of hippocampal β-catenin, BDNF, and doublecortin were significantly decreased, and the number of activated microglia increased, accompanied by amyloid-β accumulation, synaptic dysfunction, and neuroinflammation in Wnt10a^-/-^ mice. These results provided evidence that Wnt10a deficiency caused hippocampal neurodegeneration, possibly through the β-catenin signaling pathway, providing a novel insight into preventive approaches for hippocampus-dependent cognitive impairment.

## Figures and Tables

**Figure 1 biomedicines-10-01500-f001:**
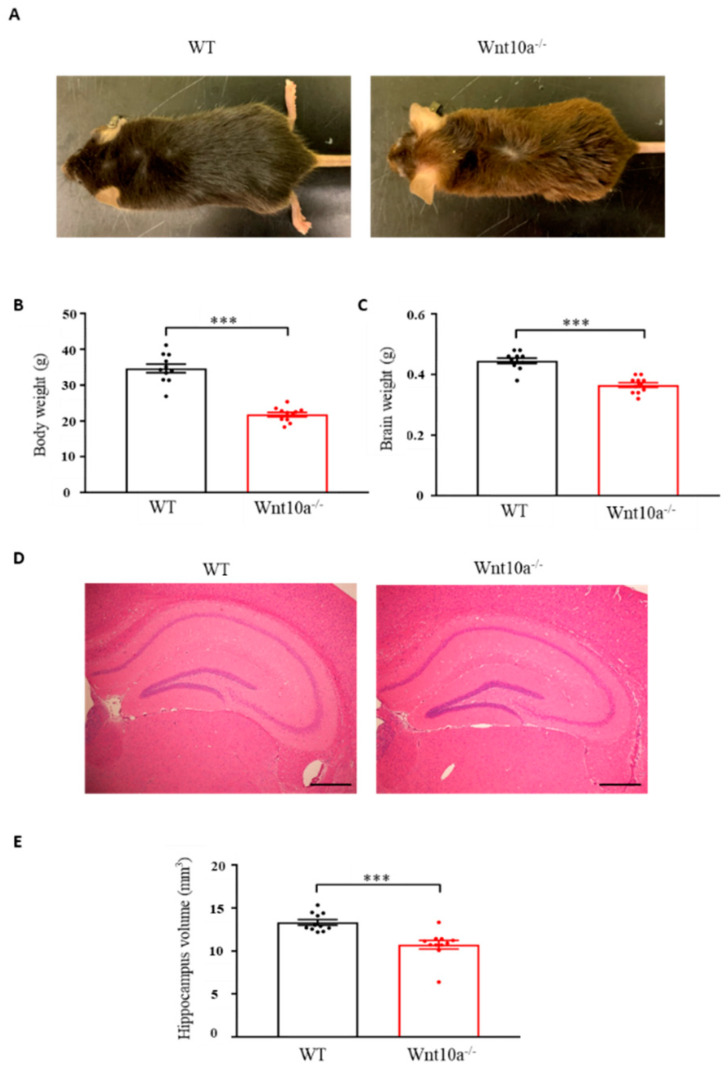
Mouse appearance, body weight, brain weight, and hippocampal volume. Compared with WT mice, Wnt10a^-/-^ mice were smaller in size (**A**). Body weights (**B**), brain weights (**C**), and hippocampal volumes (**D**,**E**) were significantly lower in Wnt10a^-/-^ mice. Data are expressed as mean ± SEM; *n* = 11/group; *** *p* < 0.001; scale bars = 500 μm.

**Figure 2 biomedicines-10-01500-f002:**
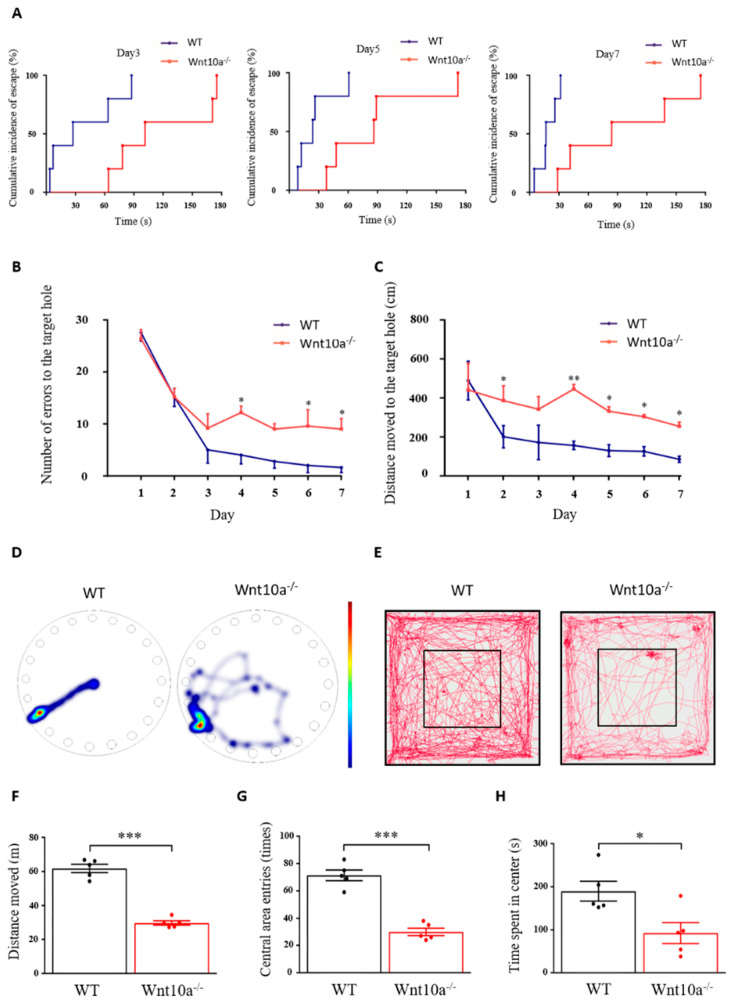
Spatial memory and behavior in Barnes maze and open field tests. The cumulative incidence of latency was longer in Wnt10a^-/-^ mice compared with WT mice (**A**). The number of errors to the target hole (**B**) was significantly higher, and the distance moved to the target hole (**C**) was significantly increased in Wnt10a^-/-^ mice compared with WT mice. Representative heatmap images showing tracking data on the final trial day (**D**) and representative movement tracks in the open field test (**E**). The total distance moved, the central area entries, and the time spent in the center were significantly decreased in Wnt10a^-/-^ mice compared with WT mice (**F**–**H**). Data are expressed as mean ± SEM; *n* = 5/group; * *p* < 0.05, ** *p* < 0.01, and *** *p* < 0.001.

**Figure 3 biomedicines-10-01500-f003:**
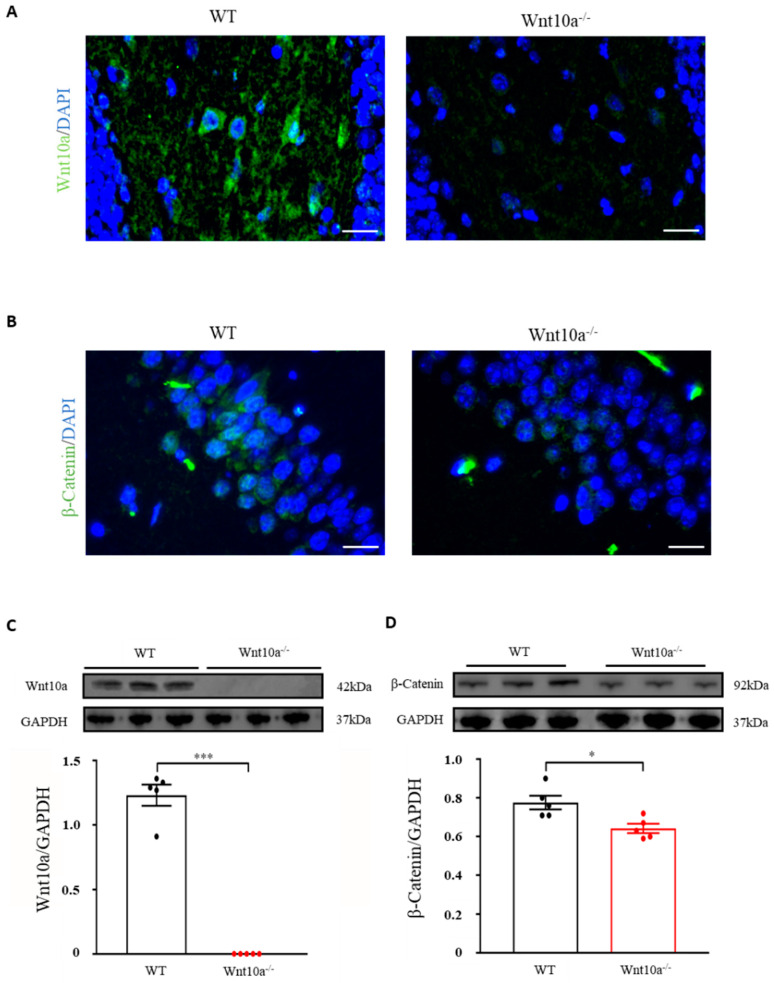
Immunofluorescence images and protein expressions of Wnt10a and β-catenin in the hippocampus. Wnt10a-positive cells were present in the hippocampi of WT mice. There no Wnt10a-positive cells in Wnt10a^-/-^ mice (**A**). The protein expression of Wnt10a was not detected in Wnt10a^-/-^ mouse hippocampi (**C**). Compared with WT mice, β-catenin-positive cells and protein expression were significantly decreased in Wnt10a^-/-^ mouse hippocampi (**B**,**D**). Scale bars = 20 μm in (**A**) and (**B**); data are expressed as mean ± SEM; *n* = 5/group; * *p* < 0.05 and *** *p* < 0.001.

**Figure 4 biomedicines-10-01500-f004:**
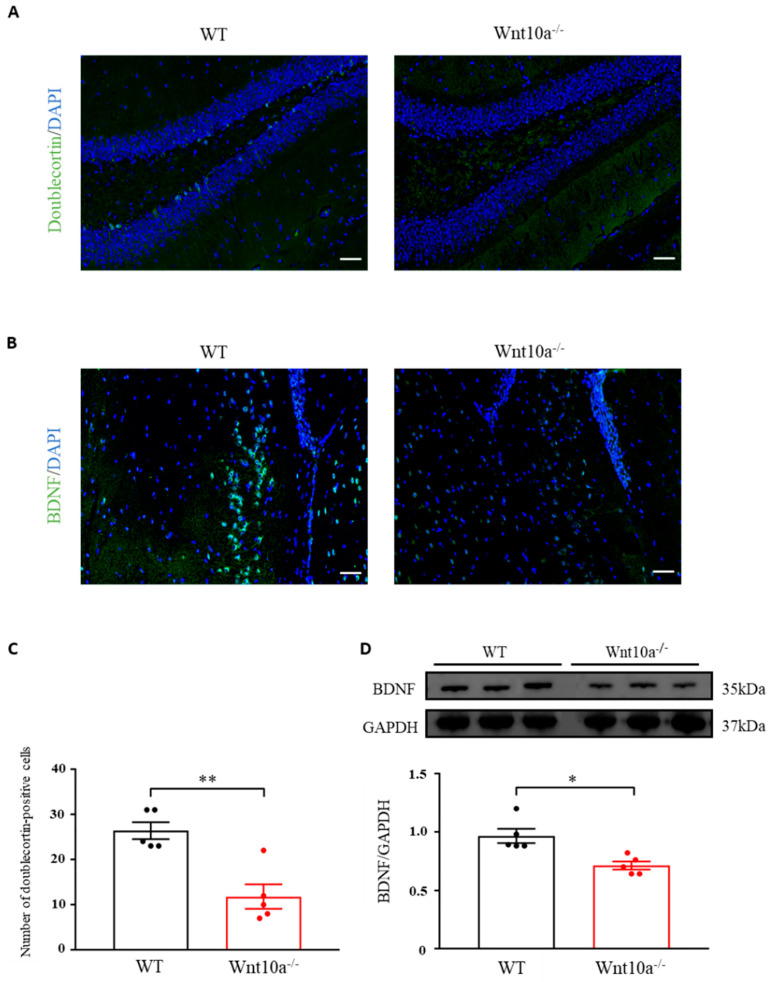
Immunofluorescence images of doublecortin and BDNF and the protein expression of BDNF in the hippocampus. The number of doublecortin-positive cells in the hippocampal dentate gyrus was significantly lower in Wnt10a^-/-^ mice compared with WT mice (**A**,**C**). The expression of hippocampal BDNF was significantly lower than that of WT mice (**B**,**D**). Scale bars = 50 μm in (**A**) and (**B**); data are expressed as mean ± SEM; *n* = 5/group; * *p* < 0.05 and ** *p* < 0.01.

**Figure 5 biomedicines-10-01500-f005:**
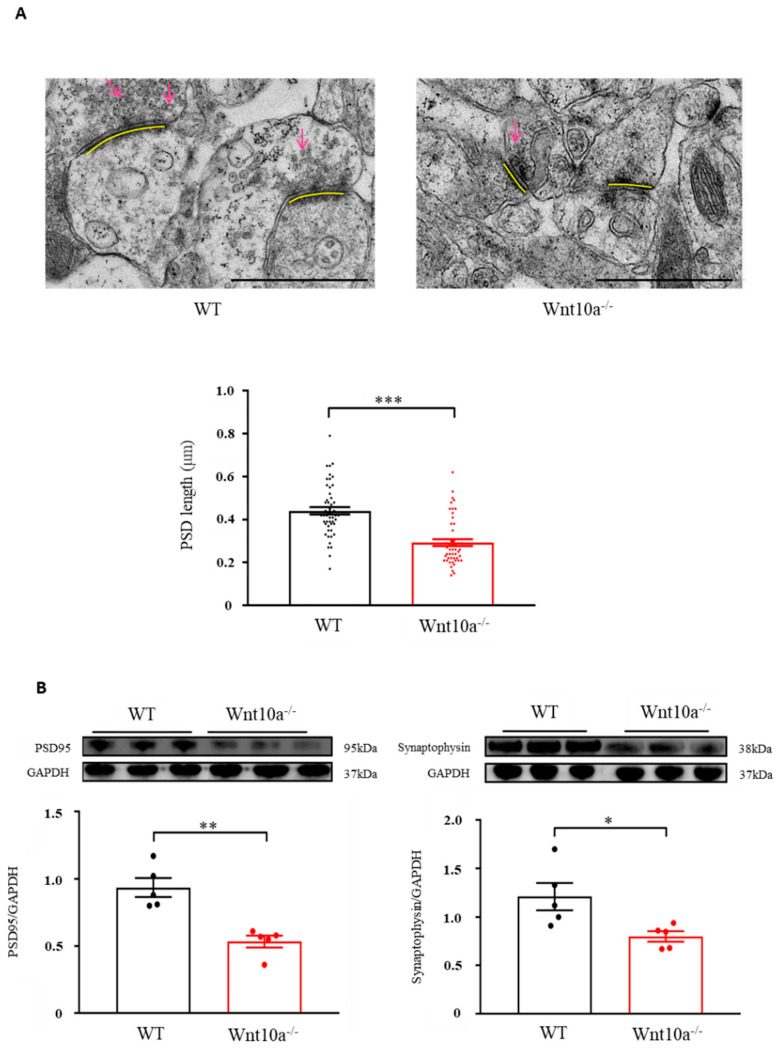
Electron microscopic images of synapses and the protein expression of PSD95 and synaptophysin in the hippocampus. The presynaptic vesicles were decreased, and the PSD length was significantly shorter in Wnt10a^-/-^ mice compared with WT mice (**A**). Arrows: presynaptic vesicles; yellow bars: PDS. Scale bars = 1 μm. The protein expressions of PSD95 and synaptophysin were significantly lower in Wnt10a^-/-^ mice than in WT mice (**B**). Data are expressed as mean ± SEM; *n* = 5/group; * *p* < 0.05, ** *p* < 0.01, and *** *p* < 0.001.

**Figure 6 biomedicines-10-01500-f006:**
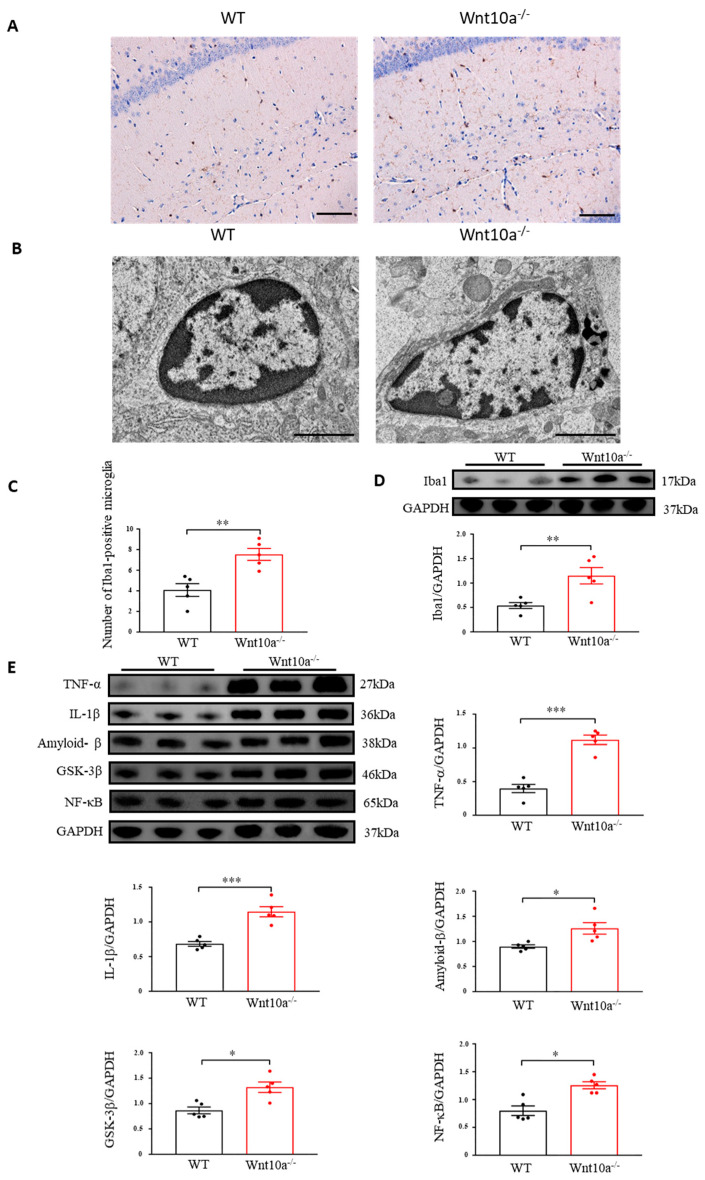
Immunostaining and electron microscopic images of microglia and the protein expression of neuroinflammatory mediators in the hippocampus. Hippocampal Iba1-positve microglia was significantly increased (**A**,**C**). Lysosomes were abundant in the hippocampal microglia of Wnt10a^-/-^ mice (**B**). Scale bars = 100 μm in (**A**) and 2 μm in (**B**). The protein expression of Iba1 was significantly higher in the hippocampi of Wnt10a^-/-^ mice (**D**). The protein expressions of hippocampal amyloid-β, NF-κB, GSK-3β, TNF-α, and IL-1β were significantly higher in Wnt10a^-/-^ mice than in WT mice (**E**). Data are expressed as mean ± SEM; *n* = 5/group; * *p* < 0.05, ** *p* < 0.01, and *** *p* < 0.001.

**Table 1 biomedicines-10-01500-t001:** Antibodies used in this study.

Antibody	CAT#	Source	MW (kDa)	Dilution	Application
Amyloid-β	Ab201060	Abcam	38	1:1000	WB
β-Catenin	Ab16051	Abcam	92	1:1000	WB/IF
Wnt10a	*		46	1:1000/1:5000	WB/IF
Doublecortin	Ab18732	Abcam		1:1000	IF
BDNF	Ab108319	Abcam	35	1:1000	WB/IF
Iba1	013-27691	Fujifilm		1:1000	IHC
Iba1	Ab48004	Abcam	17	1:1000	WB
PSD95	#2507	CST	95	1:1000	WB
Synaptophysin	#36406	CST	38	1:1000	WB
NF-κB	#8242	CST	65	1:1000	WB
GSK-3β	#9832	CST	46	1:1000	WB
IL-1β	#12426	CST	36	1:1000	WB
GAPDH	#2118	CST	37	1:1000	WB
TNF-α	#L1120	SCBT	27	1:500	WB

MW: molecular weight; WB: Western blot; IF: immunofluorescence; IHC: immunohistochemistry; CST: Cell Signaling Technology; SCBT: Santa Cruz Biotechnology. *, anti-Wnt10a antibody was produced in rabbits immunized with a synthetic peptide corresponding to amino acids 160–172 of mouse Wnt10a.

## Data Availability

Data contained in the article and the original data that support the findings of the present study are available from the corresponding author upon reasonable request.

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
