# Peer review of "Deletion of Wnt10a Is Implicated in Hippocampal Neurodegeneration in Mice"

_biomedicines, 2022, doi:10.3390/biomedicines10071500_

Round 1
Reviewer 1 Report
The manuscript by Zhang et al., describes a newly-created Wnt10a knockout mouse. The authors report significant spatial learning and memory impairments and anxiety, that occur alongside several histological changes in neurons and microglia. Several addressable concerns below:
Major
1) The introduction is a bit short and does not discuss the role of microglia. It’s unclear why AD is being singled out, many neurological diseases disproportionately affect the hippocampus; is there evidence that Wnt10a is involved in the pathogenesis of AD?
2) Mice are physically smaller, so it makes sense that their brains and hippocampi are smaller; did the authors adjust the hippocampal volumes for body weight? Are there still significant differences?
3) It would be helpful to see the distribution of all the data as well, please share individual data points in all the bar graphs.
4) Was the behavior and the subsequent histology done on the same animals? If so, do the authors have the ability to do within-animal comparisons to validate the relevance of their histological findings for behavior?
5) For the Barnes maze, escape latency is right-censored and therefore not normally distributed, rendering an ANOVA inappropriate; these data should be analyzed with a time to event (e.g. Cox proportional hazards) analysis and shown with a survival plot.
6) Can the authors also include the number of target hole checks, number of incorrect holes checks, and the primary hole distance (location of the first hole checked relative to the escape tunnel) in the Barnes maze experiment? These are good indicators of memory and spatial impairments and give a fuller indication of the impairment.
7) For open field data, the time spent in each compartment (center vs border vs corner) is not provided. The number of entries is a good indicator of anxiety; however, it is biased by distance traveled. Please include the time spent in center vs surround.
8) How was the IHC quantified (e.g. was there a fluorescence cut off for a “positive” vs “negative” cell in Figure 4?) Were investigators blinded?
9) In figure 5A, please add arrows on the figure to the presynaptic vesicles and PSD being compared for better visualization.
10) Can images in figure 6A and B be quantified?
11) Figure 6 legend description is missing (D)
12) Were there any sex effects noted?
Minor
1) Change the wording in line 42, the way the sentence is written sounds as if the new neurons are called neurogenesis rather than the process.
2) Line 43: remove “the”
3) Change hours to hour in line 99
4) Please provide the concentrations of the protein loaded into each lane of the the western blots
5) In figure 6D, one GAPDH is used, are all the proteins detected from one gel or multiple gels?
6) Line 271, involved should be involving
7) Line 294: “The synapse”
8) A few other typos remain
Author Response
Responses to the comments of Reviewer 1
We thank this reviewer for the valuable comments. The response to the comments is as follows.
Major
- The introduction is a bit short and does not discuss the role of microglia. It’s unclear why AD is being singled out, many neurological diseases disproportionately affect the hippocampus; is there evidence that Wnt10a is involved in the pathogenesis of AD?
Response: We added the following sentences in the Introduction section.
The microglia are the main innate immune cells in the brain and play a critical rule in the pathological process of AD. The dysregulated microglia release various pro-inflammatory cytokines, exacerbate amyloid-b accumulation, synaptic dysfunction, and cognitive deficits in AD progression (Frontiers in Cellular Neuroscience 2021, 15: 749587 Microglia in Alzheimer’s disease: a target for therapeutic intervention).
Impaired Wnt signaling pathway is associated with enhanced neuroinflammation, increased amyloid-b aggregation. The dysfunctional Wnt signaling might be a key event contributing to the pathogenesis of AD (Journal of Molecular Neuroscience (2022) 72:679–690).
- Compared with the WT mice, both body weights and hippocampal volumes were significantly lower in Wnt 10a-/- By adjusting the hippocampal volumes for body weight, there were no significant differences between WT and Wnt 10a-/- mice.
Response: We thank the reviewer for raising this point. Both body weights and hippocampal volumes are significantly lower in Wnt 10a-/- mice. There is no significant difference between WT and Wnt 10a-/- mice, when adjusting the hippocampal volumes for body weights.
- It would be helpful to see the distribution of all the data as well, please share individual data points in all the bar graphs.
Response: We revised all bar graphs including individual data points, according to the reviewer’s suggestion.
- Was the behavior and the subsequent histology done on the same animals? If so, do the authors have the ability to do within-animal comparisons to validate the relevance of their histological findings for behavior?
Response: We performed the behavioral and histological experiments separately. Therefore we cannot do within-animal comparison to validate the relevance of the histological findings for behavioral data.
- For the Barnes maze, escape latency is right-censored and therefore not normally distributed, rendering an ANOVA inappropriate; these data should be analyzed with a time to event (e.g. Cox proportional hazards) analysis and shown with a survival plot.
Response: We appreciate this reviewer’s thoughtful comments. The use of ANOVA in time to event data is problematic because of the right-censored nature of survival times. In our Barnes maze test, following sufficient acquisition training, animals went through the reference memory phase during the trial. All mice enter the escape hole within the 3 min period. We first performed a normal distribution test of the Barnes maze data and a homogeneity test between the groups, and then analyzed by two-way repeated measures ANOVA.
- Can the authors also include the number of target hole checks, number of incorrect holes checks, and the primary hole distance (location of the first hole checked relative to the escape tunnel) in the Barnes maze experiment? These are good indicators of memory and spatial impairments and give a fuller indication of the impairment.
Response: We thank the reviewer for this indication. We added the number of errors to the target hole and the distance moved to target hole in the sections of Materials and methods, and Results.
- For open field data, the time spent in each compartment (center vs border vs corner) is not provided. The number of entries is a good indicator of anxiety; however, it is biased by distance traveled. Please include the time spent in center vs surround.
Response: We thank the reviewer for the constructive comments. We added the time spent in the center of the box, in the sections of Materials and methods, and Results.
- How was the IHC quantified (e.g. was there a fluorescence cut off for a “positive” vs “negative” cell in Figure 4?) Were investigators blinded?
Response: We thank the reviewer for this indication. For immunohistochemical quantification, the number of doublecortin positive cells was counted in 5 sections per mouse of both groups. In each section, 10 randomly selected fields per section, original magnification ×400, were assessed and the mean number for each of the 5 mice were calculated. All measurements were conducted in a randomized double-blind procedure under the same conditions.
- In figure 5A, please add arrows on the figure to the presynaptic vesicles and PSD being compared for better visualization.
Response: We added arrows on Figure 5A to show the presynaptic vessels and PSD, according to the reviewer’s suggestion.
- Can images in figure 6A and B be quantified?
Response: We calculated the number of the Iba1-positive microglia in the section of Materials and methods. We also added a bar graph showing the number of Iba1-positive microglia, as figure 6C.
- Figure 6 legend description is missing (D) 
Response: We added (D) in Figure 6 legend description.
- Were there any sex effects noted? 
Response: It is interesting to analyze the sex effects. In this study, we just examined the alterations of male mice. We would like to investigate whether there are any sex effects in the near future.
Minor
- Change the wording in line 42, the way the sentence is written sounds as if the new neurons are called neurogenesis rather than the process.
Response: We revised the sentence as “The hippocampal DG can generate new neurons throughout life.”.
- Line 43: remove “the”
Response: Thanks. We removed “the”.
- Change hours to hour in line 99
Response: Thanks. We changed the word from “hours” to “hour”.
- Please provide the concentrations of the protein loaded into each lane of the western blots.
Response: We adjusted the protein concentration to 2.0 mg/mL for western blots.
- In figure 6D, one GAPDH is used, are all the proteins detected from one gel or multiple gels?
Response: In figure 6D, five proteins detected from one gel.
- Line 271, involved should be involving
Response: Thanks. We changed the word from “involved” to “involving”.
- Line 294: “Thesynapse”
Response: Thanks.
- A few other typos remain
Response: We thoroughly corrected typographical errors.
Reviewer 2 Report
In the introduction, it is not explained why the authors are interested in the Wnt family and why Wnt10a from about 20 Wnt proteins. Have researchers carried out any previous research in this field, or does there appear to be any evidence that this pathway plays a key role? It's not explained. The authors already have experience in this field, please indicate what previous conclusions they have at this point, as an introduction to this research.
Chapter 2
The title of chapter 2.1 "Wnt10a -/- mouse" is not precise, WT mice are also described.
"Wnt10a-/- mouse" --> "Wnt10a-/- mice"
The authors did not describe how Wnt10a - /- mice were obtained, they refer to previous work. It is worth briefly describing the procedure, then referring to the detailed information for interested readers.
There is a lack of information about the size of groups and the sex of animals. There is no information about the number of experiments and the division of animals into study groups. Were the same mice used for all behavioral studies and for the tissue testing? Has a gender effect on the results been statistically excluded?
Chapter 2.2
Behavioral experiments were conducted in the light phase. This seems inappropriate, the natural activity of rodents is at night while they sleep during the day. The daily cycle should be reversed and all research conducted during the day, in the dark phase for the rodents. It is still not described what groups of animals were tested, it is only information that there were 5 animals in the group.
Line 99
"One hours" -> "One hour"
Chapter 2.3, 2.4, 2.5, 2.6
There is still no information from which animals the tissues were collected. The division into groups and the number of individuals in the group are still unknown.
Chapter 3.2
Behavioral studies indicate worse memory, velocity, and distance shortening. How did the authors find out that the deteriorated memory was not due to sedation or muscle weakness? Administration of diazepam to mice will also result in deterioration in the memory test and shorter distance, but this is not related to memory impairment but to sedation and muscle relaxation. Therefore, authors should convince the reader that the conclusions drawn from the observations are correct and certain.
Modification of Wnt10a -/- also plays a role in peripheral changes, outside the nervous system - tissue injury/wounds and subsequent tissue repair and fibrosis, i.e., wound healing/scarring, along with the production of extracellular matrix (ECM) including collagen, through mechanisms could be involved in WNT signaling. It should be assumed that also the malfunctioning of the organism caused by the exclusion of Wnt10a may contribute to the poorer performance of mice and result directly in lower body weight and poor behavioral functioning.
The discussion is written correctly and very interesting.
Author Response
Responses to the comments of Reviewer 2
We thank this reviewer for the valuable comments. The response to the comments is as follows.
- In the introduction, it is not explained why the authors are interested in the Wnt family and why Wnt10a from about 20 Wnt proteins. Have researchers carried out any previous research in this field, or does there appear to be any evidence that this pathway plays a key role? It's not explained. The authors already have experience in this field, please indicate what previous conclusions they have at this point, as an introduction to this research.
Response: In mammals, there are 19 different Wnts. Impaired Wnt signaling pathway, including Wnt1, Wnt3a, Wnt5a, Wnt7a and Wnt7b, is associated with enhanced neuroinflammation, increased amyloid-b aggregation. The dysfunctional Wnt signaling might be a key event contributing to the pathogenesis of AD (Journal of Molecular Neuroscience (2022) 72:679–690). No reports, however, have examined the effects of Wnt10a on hippocampal structure and function. In this study, we investigated the effects of Wnt10a on hippocampus-dependent cognition, using Wnt10a-/- mice.
- The title of chapter 2.1 "Wnt10a -/- mouse" is not precise, WT mice are also described.
"Wnt10a-/- mouse" --> "Wnt10a-/- mice"
Response: We changed the word "Wnt10a -/- mouse" to "Animals".
- The authors did not describe how Wnt10a - /- mice were obtained, they refer to previous work. It is worth briefly describing the procedure, then referring to the detailed information for interested readers.
Response: We thank the reviewer for this indication. We added the following sentences in chapter 2.1.
Briefly, we obtained the embryonic stem cells carrying the deleted 12663 base pairs of mouse chromosome 1, which included the entire Wnt 10a coding region from the University of California, Davis, School of Veterinary Medicine (Davis, CA, USA). The embryonic stem cell clones were microinjected into the blastocysts of C57BL/6J mice to generate chimeric mice. The successful deletion of Wnt 10a was determined by genotyping PCR method.
- There is a lack of information about the size of groups and the sex of animals. There is no information about the number of experiments and the division of animals into study groups. Were the same mice used for all behavioral studies and for the tissue testing? Has a gender effect on the results been statistically excluded?
Response: As to the size of groups, we used 11 mice for histological observation, 5 mice for immunostaining, western blot, behavioral test, and electron microscopy, respectively. As to the sex of animals, in this study we just examined the alterations of male mice. We would like to investigate whether there are any sex effects in the near future.
- Chapter 2.2
Behavioral experiments were conducted in the light phase. This seems inappropriate, the natural activity of rodents is at night while they sleep during the day. The daily cycle should be reversed and all research conducted during the day, in the dark phase for the rodents. It is still not described what groups of animals were tested, it is only information that there were 5 animals in the group.
Response: We thank the reviewer for this indication. The fact that mice are nocturnal animals and more active at night. The behavioral experiments carried out in the light phase should consider the possible effects of the testing time and light condition. The previous studies showed that the circadian rhythm affects the behavioral activity (Lab Anim. 40:371-381, 2006). However, some studies demonstrated that by comparing mouse behavioral experiments in the light and dark phases, no significant differences were found regarding the behavioral parameters between night and daytime testing (Mol. Psychiatry 20:1479-1480, 2015; Scientific Reports 12:432, 2022). Therefore, it is necessary to take the diurnal rhythm into account for behavioral experiments.
- Line 99 "One hours" -> "One hour"
Response: Thanks. We corrected the word.
- Chapter 2.3, 2.4, 2.5, 2.6
There is still no information from which animals the tissues were collected. The division into groups and the number of individuals in the group are still unknown.
- Response: We added the animal numbers in Chapter 2.3, 2.4, 2.5, 2.6, respectively.
- Chapter 3.2
Behavioral studies indicate worse memory, velocity, and distance shortening. How did the authors find out that the deteriorated memory was not due to sedation or muscle weakness? Administration of diazepam to mice will also result in deterioration in the memory test and shorter distance, but this is not related to memory impairment but to sedation and muscle relaxation. Therefore, authors should convince the reader that the conclusions drawn from the observations are correct and certain.
Response: Thank you for your indication. Sedation or muscle weakness could affect memory. We did not use any sedative agents in this behavioral experiment. The body weights of Wnt10a-/- mice was significantly lower than that of the WT mice. There were no significant differences regarding the muscle weights between Wnt10a-/- and WT mice (Bone 120:75-84, 2019). The effects of muscle weakness on worse memory need to be confirmed in the near future.
- Modification of Wnt10a -/- also plays a role in peripheral changes, outside the nervous system - tissue injury/wounds and subsequent tissue repair and fibrosis, i.e., wound healing/scarring, along with the production of extracellular matrix (ECM) including collagen, through mechanisms could be involved in WNT signaling. It should be assumed that also the malfunctioning of the organism caused by the exclusion of Wnt10a may contribute to the poorer performance of mice and result directly in lower body weight and poor behavioral functioning.
Response: We thank this reviewer for the constructive comments. Wnt10a-/- mice exhibit multiple alterations, including lower body weight, wound healing, osteogenesis, and adipogenesis. Further studies are necessary to clarify whether these factors influence hippocampus-dependent cognition in Wnt10a-/- mice.
Round 2
Reviewer 1 Report
The authors have addressed most of my concerns. A more thorough exploration of the behavioral data greatly strengthens the manuscript. Future studies should include female mice. Two outstanding issues remains:
1) "There is no significant difference between WT and Wnt 10a-/- mice, when adjusting the hippocampal volumes for body weights." Please include this statement in the result section of the manuscript.
2) The ANOVA is really not appropriate for latency data, not only because it is right censored. Please see: https://www.ncbi.nlm.nih.gov/pmc/articles/PMC3103828/
Author Response
Responses to the comments of Reviewer 1
We thank this reviewer for the valuable comments. The response to the comments is as follows.
1) "There is no significant difference between WT and Wnt 10a-/- mice, when adjusting the hippocampal volumes for body weights." Please include this statement in the result section of the manuscript.
Response: We added this statement in the result section, as the reviewer indicated.
2) The ANOVA is really not appropriate for latency data, not only because it is right censored. Please see: https://www.ncbi.nlm.nih.gov/pmc/articles/PMC3103828/
Response: We thank the reviewer for raising this point. We analyzed the cumulative incidence of escape and the hazard ratio using Log-rank (Mantel-Cox) test and Mantel-Haenszel method, instead of ANOVA.
